# Coevolutionary methods enable robust design of modular repressors by reestablishing intra-protein interactions

Xian-Li Jiang[1,2,8], Rey P. Dimas[3,8], Clement T. Y. Chan [4,5✉] & Faruck Morcos [1,6,7✉]

Genetic sensors with unique combinations of DNA recognition and allosteric response can be created by hybridizing DNA-binding modules (DBMs) and ligand-binding modules (LBMs) from distinct transcriptional repressors. This module swapping approach is limited by incompatibility between DBMs and LBMs from different proteins, due to the loss of critical module-module interactions after hybridization. We determine a design strategy for restoring key interactions between DBMs and LBMs by using a computational model informed by coevolutionary traits in the LacI family. This model predicts the influence of proposed mutations on protein structure and function, quantifying the feasibility of each mutation for rescuing hybrid repressors. We accurately predict which hybrid repressors can be rescued by mutating residues to reinstall relevant module-module interactions. Experimental results confirm that dynamic ranges of gene expression induction were improved significantly in these mutants. This approach enhances the molecular and mechanistic understanding of LacI family proteins, and advances the ability to design modular genetic parts.

[1] Department of Biological Sciences, The University of Texas at Dallas, Dallas, TX, USA. [2] Department of Bioinformatics and Computational Biology, The University of Texas M.D. Anderson Cancer Center, Houston, TX, USA. [3] Department of Biology, The University of Texas at Tyler, Tyler, TX, USA. [4] Department of Biomedical Engineering, University of North Texas, Denton, TX, USA. [5] BioDiscovery Institute, University of North Texas, Denton, TX, USA. [6] Department of Bioengineering, The University of Texas at Dallas, Dallas, TX, USA. [7] Center for Systems Biology, The University of Texas at Dallas, Dallas, TX, USA. [8] These authors contributed equally: Xian-Li Jiang, Rey P. Dimas. ✉email: tszyanclement.chan@unt.edu; faruckm@utdallas.edu

Synthetic genetic circuit approaches represent programmable ways to create distinct signal response behavior in cells and organisms[1–3]. However, the construction of genetic circuits is constrained by the lack of modular components; in natural biological systems, a biological sensor usually responds to a unique molecular signal to control gene expression driven by a specific genetic element, such as a promoter. This rigidity poses a challenge for scientists to implement diverse circuit designs in biological systems[4–7].

To address this problem, a module swapping strategy has been used to create modular biosensors with allosterically regulated transcription repressors. We recently demonstrated that repressors in the LacI and TetR families are composed of two discrete, conserved, and potentially interchangeable modules that are responsible for the detection of ligands and for interaction with DNA-based promoters. Therefore, fusing a ligand-binding module (LBM) and a DNA-binding module (DBM) from different transcription repressors generates a hybrid repressor with a unique combination of the two properties[8–10]. Other research groups have also generated hybrid repressors from the LuxR, OmpR, and NarL families[11,12].

These hybrid repressors facilitate flexible connections from chemical signals to promoters for controlling gene expression; specifically, we can harness modular repressors to implement circuit topologies that require multiple signaling molecules to activate the expression of an output gene, or to use one signal to induce gene expression driven by different promoters; these properties create further possibilities in circuit design. For example, we previously used hybrid repressors to develop a programmable 'Passcode' genetic device for bacterial biocontainment[8]. As an additional example, we have established a system of multiple toggle switches with a master OFF signal[9]. Other studies also demonstrated the use of hybrid repressors for generating Boolean logic operations[13,14]. In addition to circuit development, module swapping of repressors has been used for other applications. For instance, Schmidl et al. propose to swap the DNA-binding domain of unknown repressors to eliminate the need of characterizing DNA recognition properties, which facilitates the screening of effectors of new proteins[12]; Mukherji et al. used a hybrid repressor to regulate the expression of metabolic genes in a bacterial genome, such that a new environmental signal can be used to control natural product production[11]. The module swapping strategy positively impacts bioengineering in a broad range of directions.

These hybrid repressors are powerful for engineering cellular behavior. However, one major challenge in creating hybrid repressors is that some DBMs and LBMs are incompatible, as supported by our recent publications[9,10]. This incompatibility leads to poor performance of some hybrid repressors, which impedes the linkage between many inputs and outputs. Other studies also support that this is a common problem in engineering repressors with the module swapping strategy, as some hybrid repressors generated from members of the LuxR, OmpR, and NarL families are less efficient comparing to the native repressors[11,12]. Uncertainties in designing hybrid repressors may hinder this strategy for applications that require the use of specific LBMs and DBMs. To overcome this problem, it is necessary to enhance the robustness of the design and construction of modular repressors.

The aim of this study is to establish a robust strategy for modifying poorly functional hybrid repressors, which rescues their activities. A major strategy in protein engineering involves using computational algorithms to design huge pools of mutants and then using high-throughput screening to experimentally identify desirable candidates[15–17]. This strategy requires a large amount of resources for rescuing each protein and thus, it may not be suitable for routine engineering of hybrid repressors. Addressing our problem requires a knowledge-based method that is applicable to different hybrid repressors, allowing the identification of specific mutations for improving protein activities. To this end, we explored a rational design approach that is based on coevolutionary traits within the protein family.

We hypothesized that the loss of critical module-module interactions is a major cause of reduced protein activities in hybrid repressors. To test this hypothesis, our overall approach is to predict mutations that may restore native-like interactions interrupted by the hybridization process between different repressors. We previously developed a computational model to study interactions between LBMs and DBMs[9]; this model is based on the assumption that a network of evolutionary relevant DBM-LBM couplings are essential for allosteric protein function and thus, these interactions are highly coevolving during the history of the family. As a result, if an amino acid residue involved in such a network is changed during evolution, its interacting residue needs to coevolve to maintain its function. Identification of interacting proteins and domains via iterative optimization of evolutionary couplings has been successful at predicting cognate interactions in two-component systems[18–21], providing conceptual support to investigate compatibility among independently folded units in repressors. We, therefore, analyzed over 70,000 LacI homologs to identify inter-modular residue pairs that coevolve and then utilized these pairs to compute a compatibility score $C(\mathbf{S})$ where $\mathbf{S}$ represents a given hybrid protein sequence that consists of different LBMs and DBMs, which successfully inferred the performance of hybrid repressors.

In this work, we use this coevolutionary modeling approach to engineer mutations within LBMs that can restore desirable module-module interactions for rescuing hybrid repressor activities. Our results also suggest that some residues involved in module-module interactions should not be altered because they also play key roles in interacting with other residues within the LBM to maintain structural integrity. Based upon these discoveries, we establish a computational approach to guide the engineering of eight hybrid repressors with poor activities into functional hybrids. This model accurately predicts which hybrids can be rescued by mutating residues at the module-module interface and according to these proposed changes, we successfully construct mutants with significantly improved performance. Overall, our work demonstrates the power of coevolutionary analysis in the process of protein engineering and showcases potential applications in understanding molecular communication and synthetic biology.

## Results

**Coevolutionary cues help reestablish module-module interactions in hybrid repressors.** In this study, we harnessed our DBM-LBM compatibility model to predict mutations that are expected to improve the functionality of hybrid repressors. A compatibility score $C(\mathbf{S})$ for a hybrid repressor was computed using the inter-modular coevolutionary coupling strength parameters, $e_{ij}(A_i, A_j)$, inferred from multiple sequence alignment of LacI homologs using global inference of the joint distribution of sequences in the family[22] (see Online Methods). For a given hybrid sequence, a mutation at residue $i$ updates all parameters, $e_{ij}(A_i, A_j)$, which describe interactions of the mutated residue, $i$, with all of its coevolving partners, $j$, resulting in a change in $C(\mathbf{S})$ score (Fig. 1a). Using this computational model, we systematically computed $C(\mathbf{S})$ scores for mutations at LBMs. We did not consider mutations at DBMs because these modules are small (approximately 47 amino acid residues) and many residues are directly involved in DNA-binding and recognition[23]; mutating a

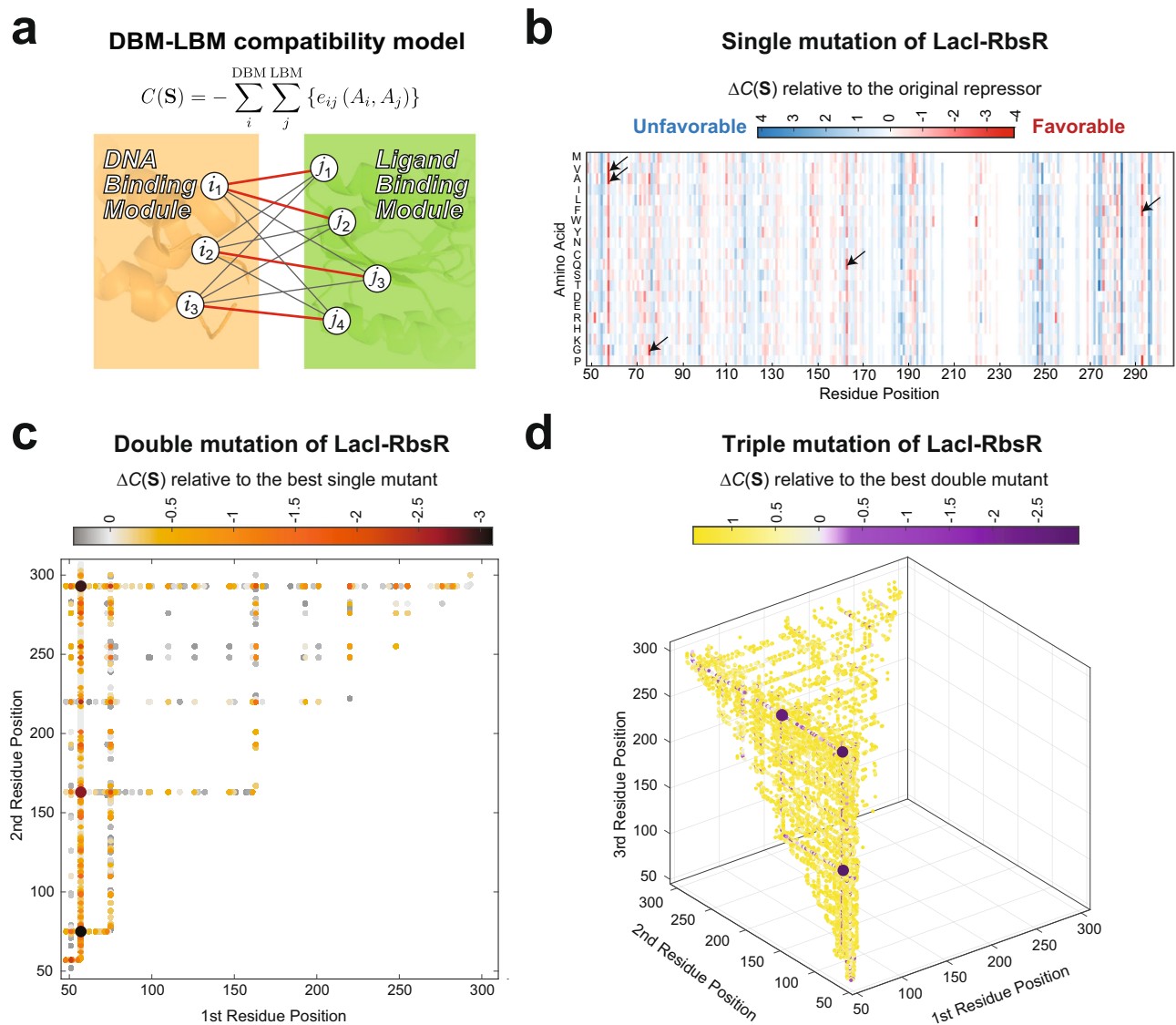

**Fig. 1 A computational approach to modulate hybrid repressors. a** We identify a set of DNA-binding modules (DBMs) and ligand-binding modules (LBMs) interactions based on coevolutionary cues among LacI family members. These interacting residue pairs were used to develop a computational model for computing a compatibility score $C(\mathbf{S})$ that reflects compatibility between a DBM and an LBM. **b** We then applied this model to improve the repressor activities of a hybrid repressor, LacI-RbsR. The heatmap shows the $\Delta C(\mathbf{S})$ scores of all LacI-RbsR candidates with a single mutation at its LBM relative to the score for original repressor ($C(\mathbf{S})$ score: −62.10). Similarly, we calculated the $C(\mathbf{S})$ scores of (**c**) double mutants and (**d**) triple mutants and plotted the $\Delta C(\mathbf{S})$ scores relative to the best single mutant and best double mutant, respectively.

DBM is likely to affect the DNA-binding properties of the protein. We then selected mutants with the best $C(\mathbf{S})$ scores, which represent candidates for improving repressor activities.

To test whether our approach can rescue hybrid repressors, we applied it to 8 hybrid repressors. From our previous study[9], we generated a total of 35 hybrid repressors; 18 of them are highly efficient and produce a dynamic range of induction over 10-fold; another 8 hybrids are poorly functional, which generate significantly reduced dynamic range of induction (3- to 10-fold); and the last 9 hybrids have no significant activities (induction is less than 3-fold). For the 8 cases with reduced activities, the fact that these hybrids still have some biological activity suggests that the LBM and the DBM can still interact to some extent; however, for those other 9 hybrids with no activities, all module-module interactions could be completely lost. We then selected those 8 repressors with reduced activities because they may represent cases where a majority of essential LBM-DBM interactions are

maintained and only a few interacting pairs are disrupted; therefore, only a few mutations may be necessary to fully restore repressor activities. We first analyzed a hybrid repressor, LacI-RbsR, from our previous study, which generates a 5-fold increase of induction in response to its inducer, ribose. This hybrid contains a LacI DBM and an RbsR LBM (all hybrids in this study are named following this pattern). A heatmap is shown in Fig. 1b to illustrate the effect on the compatibility score of all possible single mutations for this hybrid repressor. Among the mutations that improve the compatibility score, the top 5 favorable mutations are K57V, K57A, F75G, N163Q, and K295F. We only considered mutations at the LBM, therefore the mutational effects on compatibility scores are additive since our model only analyzes coevolutionary cues between DBMs and LBMs. Consequently, mutants with two and three of these favorable mutations lead to a further improved compatibility score. Therefore, the top double mutations involve the same residue positions (57,75,163, and 295)

as found in the single mutation profile (Fig. 1b). In the triple mutation profile, residue positions 57, 75, 163, and 295 were also involved in the top 2 mutants (Fig. 1c).

After analyzing LacI-RbsR, we applied this approach to study other hybrid repressors with similar performance. We studied a total of 8 hybrids, which originally only generated 3- to 10-fold induction in gene expression: LacI-RbsR, PurR-GalR, CelR-RbsR, RbsR-GalR, RbsR-LacI, XltR-GalR, MalR-LacI, and XltR-ScrR (Supplementary Fig. 1). We then predicted triple mutations for each hybrid repressor that possess the best $C(\mathbf{S})$ score.

**Designing hybrid repressors to improve allosteric regulation activities.** We then tested whether our coevolutionary modeling approach is sufficient to rationally improve the performance of hybrid repressors. For all eight hybrid repressors mentioned above, two triple mutation candidates with top $C(\mathbf{S})$ scores were selected for experimental characterization. In all eight cases, the two 3-mutation candidates have two shared mutations; within the three mutations of each candidate, we considered all possible combinations with single, double, and triple mutants, which led to a set of 11 mutants for each hybrid repressor (Source Data file). The additive effect of multiple mutations on the $C(\mathbf{S})$ score ensured the single/double mutations were also among the top candidates in their respective cohort (Figs. 1b to 1d).

Each mutant was characterized with our in vivo transcriptional assay using *Escherichia coli* cells—the repressor was constitutively expressed in cells to repress the expression of GFP. Activities of allosteric response and transcriptional regulation were assessed by comparing GFP levels in cells that were exposed and unexposed to the corresponding inducer of the target repressor. Characterization data from all 88 mutants from the 8 hybrid repressors are listed in the Source Data file. Our predicted mutations significantly improved the activities of four hybrid repressors, such that they became capable to generate GFP signal inductions that were above 10 fold. These modified repressors include LacI-RbsR (Fig. 2a and Supplementary Fig. 2a), PurR-GalR (Fig. 2b and Supplementary Fig. 2b), CelR-RbsR (Fig. 2c and Supplementary Fig. 2c), and RbsR-GalR (Fig. 2d and Supplementary Fig. 2d). Among mutants with the best performance for each rescued hybrid, there are mutations at three homologous positions (Supplementary Fig. 3), in which some of them are far from the DBM-LBM interface, suggesting that residues at these positions are not directly involved in module-module interactions but they may play key roles in modulating protein confirmation at that interface for facilitating repressor function. These results also reveal the power of coevolutionary coupling analysis in discovering intra-protein interactions.

For the original version of these four hybrid repressors, the poor dynamic range of induction can be due to defects in different protein properties—the original LacI-RbsR and CelR-RbsR exhibited high uninduced expression level which indicates weak DNA-binding; in contrast, the original PurR-GalR and RbsR-GalR generated low basal expression but repression was not fully released upon induction, suggesting allosteric properties of these repressors were reduced. Intriguingly, we successfully used our model to predict mutations that restore different functions among these repressors. K57 in LacI-RbsR and K60 in CelR-RbsR are homologous positions located at the hinge helix motif (Supplementary Fig. 3) and directly contacting the backbone of DNA but not the nucleobases; it is proposed that the hinge helix is involved in facilitating DNA-protein binding but not recognizing the operator sequence[8]. Thus, our results suggest that this position plays a key role in interacting with specific groups of the DNA backbone, such that the DNA and LBM reach a desirable orientation for forming a complex. For the other two

rescued repressors, A85/A123 in PurR-GalR and A85/A123 in RbsR-GalR are distal to DNA and more likely to be involved in a role at inducing allosteric response only[24,25]. These results strongly imply that disruption of different residue pairs for DBM-LBM interactions can have a specific influence on DNA-binding and allosteric response.

On the other hand, we observed that hybrid repressors with similar functional defects can be caused by the disruption of different interacting pairs. For PurR-GalR and RbsR-GalR, the two hybrids are structurally similar as they both contain a GalR LBM. The original version of these two repressors also performed similarly, in which both bound tightly to DNA but did not release efficiently from the promoter upon induction. We first hypothesized that these two hybrid repressors had lost a homologous module-module interacting pair. However, our experimental characterization shows that different mutations are required for rescuing the two repressors. PurR-GalR needed three mutations to reach its best performance (245-fold induction), including A55V, A85C, and A123C. Using only the mutants A85C and A123C, there was only 12-fold induction in expression (Source Data file), suggesting that A55V is critical. However, RbsR-GalR gained an improvement of induction fold-change to 69 fold with only A85C and A123C, which are homologous to PurR-GalR's A85C and A123C, respectively. These results suggest that there are no universal sites that are able to rescuing repressors; they form a complex network of interactions that can only be revealed with a global model and metric like the one introduced here.

**Folding and structural constraints on highly compatible mutants.** While we rescued four hybrid repressors, the performance of another four repressors were not improved based on our model predictions. Moreover, some repressors' activities were enhanced with one or two mutations, but not with triple mutations, even though it was inferred that the triple mutant versions would have more favorable compatibility scores (Source Data file). For instance, the original RbsR-GalR generated a 3-fold induction in GFP fluorescence in response to the inducer, galactose; a single mutation A85C or two mutations, A85C and A123C, enhanced the induction to 43-fold and 69-fold, respectively; from our coevolutionary model, the compatibility scores are −69.08 for RbsR-GalR A85C and −71.93 for RbsR-GalR A85C/A123C. A third mutation on RbsR-GalR (G67T or H122M) was expected to further enhance the compatibility between the DBM of RbsR and LBM of GalR; however, the resulting mutants, G67T/A85C/H122M and A85C/H122M/A123C, were inactive, generating induced fold-change of 1.2 and 1.5, respectively. On the other hand, K57A in LacI-RbsR improved the induced fold-change level to 260-fold from an original 6-fold induction, while a double mutation containing K57A and N163Q, and a triple mutation of K57A/F75G/K295F resulted in relatively lower induction levels, 170-fold and 147-fold, respectively. These results suggest that residues at some positions may have multiple molecular roles; while our model identified that they are involved in DBM-LBM interactions, they may be critical for maintaining structure and function within the LBM, in which mutating these residues leads to a loss of protein function. Our original compatibility model only took into consideration of coevolutionary cues between inter-domain residue pairs and did not examine residue pairs within each module. Therefore, mutants designed by this model did not evaluate the structure and function of resulting LBMs.

In order to improve our model for the design of hybrid repressors with high induction, it is necessary to further understand molecular interactions within LBMs. For this purpose, we introduced an additional metric into our model,

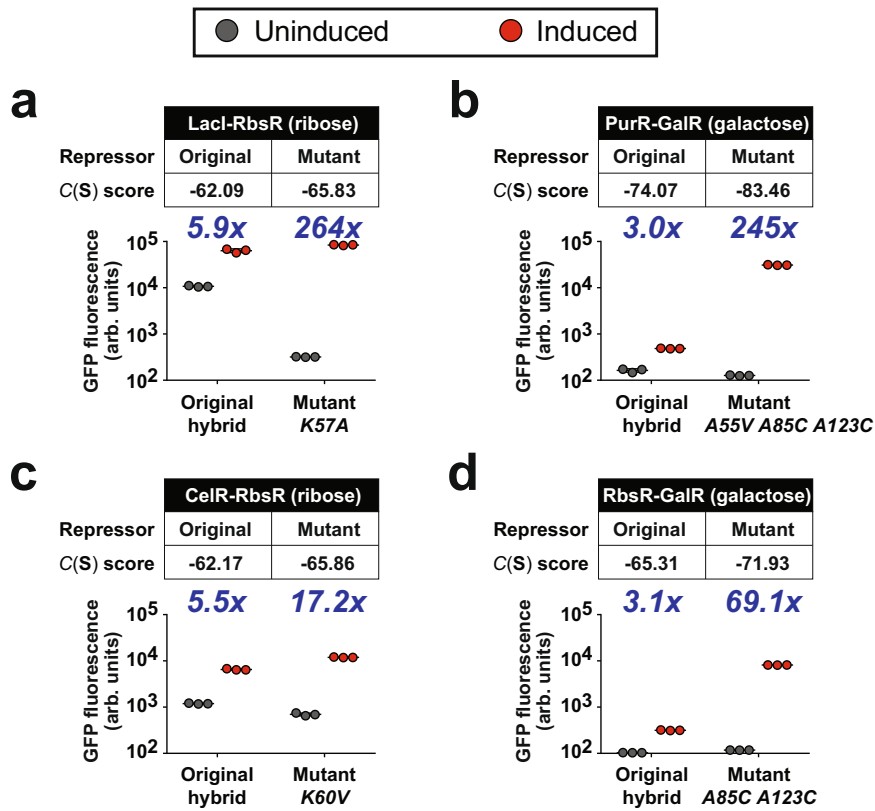

**Fig. 2 Experimental characterization of modified hybrid repressors with significant improved performance.** We assessed the ability of hybrid repressors in gene expression regulation and allosteric response, by using a transcriptional reporter assay in a strain of *E. coli* with GFP as a reporter of transcription activities. Four hybrid repressors were improved, including (**a**) LacI-RbsR, (**b**) PurR-GalR, (**c**) CelR-RbsR, and (**d**) RbsR-GalR. As shown in the table in each panel, our compatibility model predicts that each mutant gains improved performance, based on the compatibility scores *C*(**S**). Indeed, these mutants showed increases in a dynamic range of gene expression in response to their inducers. GFP fluorescence in cells measured from three biological replicates are illustrated with markers that are gray (uninduced) and red (induced). The inducer used is shown inside a bracket in each table. The blue number above each plot represents the corresponding fold-change of GFP induction. The mean ± S.D. of the three biological replicates is also shown in each plot. Source data are provided as a Source Data file.

which is based on residue proximity and coevolutionary traits within LBMs. A structure-based score, *SF*(**S**), was computed by combining coevolutionary strength between residues within the LBM, with residue-residue distance below 10 Å in a LacI X-ray crystal structure (see Online Methods). Similar to the *C*(**S**) score, a mutant with increased (more positive) *SF*(**S**) is considered as less structurally stable and it may not maintain its protein function.

We then tested whether *SF*(**S**) can serve as a selection tool to eliminate mutations that lead to a loss of repressor activities. Among all mutants of LacI-RbsR (Fig. 3a), only the K57A mutant has a more favorable *SF*(**S**) score compared to its original repressor and indeed, it had the best performance (245-fold induction). Two additional LacI-RbsR mutants are also significantly improved, including K57A/N163Q (170-fold) and K57A/F75G/K295F (147-fold), and their *SF*(**S**) scores rank number 2 and number 4, respectively.

Similarly, for PurR-GalR (Fig. 3b), CelR-RbsR (Fig. 3c), and RbsR-GalR (Fig. 3d), the mutant with the best performance has an *SF*(**S**) score better than its original protein (Source Data file). In total, 11 mutants from these four hybrid repressors have an *SF*(**S**) score better than the original, and 10 of them have fold induction improved to above 10. In contrast, among those other four hybrid repressors that have not been rescued, including, RbsR-LacI (Fig. 3e), XltR-GalR (Fig. 3f), MalR-LacI (Fig. 3g), and XltR-ScrR (Fig. 3h), a majority of their mutants have an *SF*(**S**) score worse than their original repressor, indicating that these

mutants may not be functional due to possible negative structural effects within the LBM. The few exceptions include MalR-LacI S195I, XltR-GalR A301G, and XltR-GalR E226L/A301G, which have improved *SF*(**S**) scores but they remained poorly functional; based upon the crystal structure of LacI[24], the homologous residues at these positions (S195 of MalR-LacI and A301 of XltR-GalR) are at close proximity to the ligand-binding pocket and mutating them may directly interrupt ligand-binding (Supplementary Fig. 4), which provides a plausible explanation on the poor functionality of these three mutants. To enhance the robustness of our computational tool, we could prohibit mutations in the ligand-binding pocket. In our current model, we do not consider any mutations in the DNA-binding module because many residues there are highly involved in interacting with the DNA operator. Similarly, we could eliminate all mutations that are interacting with the ligand directly. Overall, these results strongly support that the *SF*(**S**) score reliably indicates whether an LBM mutation is expected to negatively affect repressor activity.

After evaluating the capability of our *SF*(**S**) model for predicting mutant performance, we also used this model and a native LacI crystal structure to understand how some mutations may disturb intra-module interactions. Taking the case of RbsR-LacI as an example, all four mutations do not improve the repressor function and all of them lead to less favorable *SF*(**S**) scores (Fig. 3e and Source Data file). We then identified interacting partners of these four mutations from our model,

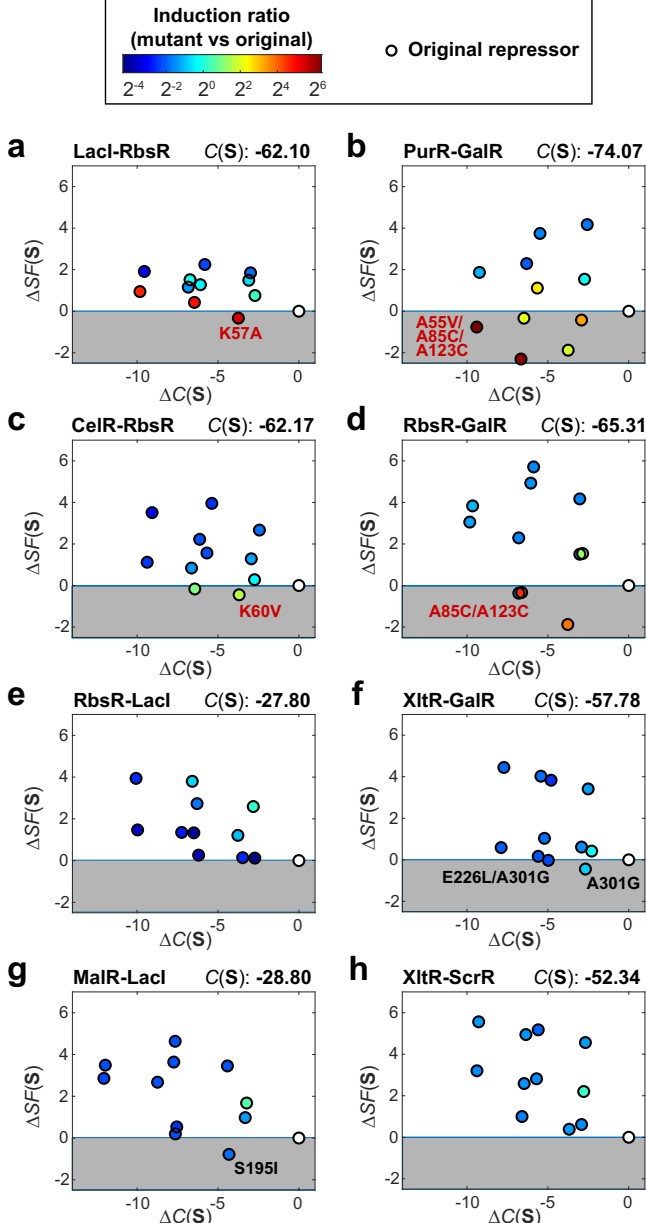

**Fig. 3 Prediction from the SF(S) score model on structural feasibility of mutations for enhancing hybrid repressor activities.** The structure-based scores, SF(S) are plotted against the compatibility scores, C(S) from experimentally tested mutant candidates of each hybrid repressors, including (**a**) LacI-RbsR (**b**) PurR-GalR, (**c**) CelR-RbsR,(**d**) RbsR-GalR, (**e**) RbsR-LacI, (**f**) XltR-GalR, (**g**) MalR-LacI and (**h**) XltR-ScrR. In these plots, mutants in the gray region have a more favorable SF(S) score compared to their original repressor. Colorbar indicates the ratio of GFP fluorescence induction for each mutant relative to the original repressor hybrid (marker filled with white), with red and blue to represent the increase and decrease of GFP fluorescence, respectively. For repressors with improved mutants (fold-change of the mutant is >10 and >3 times of the original hybrid repressor), the mutant with the largest dynamic range of induction is labeled in red. For repressors without improved mutants, the mutants are labeled in black if their SF(S) score is more favorable than their original repressor, which represents exceptional cases that do not follow our model prediction. Source data are provided as a Source Data file.

which are major contributors to the change of the SF(S) score (Supplementary Fig. 5). By studying these identified residues, we gained insights into how these mutations may affect protein function. For mutation V148W in RbsR-LacI (Supplementary Fig. 5a), valine 148 is surrounded by hydrophobic residues, such as A131 and L194; mutating V148 to a tryptophan may interrupt the hydrophobic interactions between these positions. Additionally, from the crystal structure, several polar residues are at close proximity to V148 but they face the opposite direction, including D127, S191, and S189; the mutation V148W may lead to new hydrogen bonds between the tryptophan and these polar residues, which can trigger significant change in protein confirmation. For mutation N123M (Supplementary Fig. 5b), N123 is likely to form a strong hydrogen bond with S67, which can be disrupted when mutated to a methionine. For A85C (Supplementary Fig. 5c), A85 is located at an α-helix and it interacts with residues at a neighboring β-sheet, including V92 and L60. Changing A85 to a cysteine may affect the hydrophobic interface, which is likely to destabilize the β-sheet. Finally, mutation G56V (Supplementary Fig. 5d) is located at the hinge helix, which plays an essential role in transmitting allosteric signals for controlling DNA-binding affinity. G56 interacts with two other residues on the hinge helix, R49, and A51, in which G56V may destabilize the hinge helix and affect its functionality. In addition to protein design, these examples show how we may also use the SF(S) model to facilitate the study of protein mechanisms at a molecular level.

## Discussion

Transcription repressors within the same family have structurally conserved and independently folded DNA-binding and ligand-binding domains. This fact provides the basis to support the idea of swapping these modules to create hybrids with unique combinations of signal sensing and DNA recognition properties. While module swapping is structurally feasible among family members, many resulting hybrid repressors are poorly functional, which led us to hypothesize that some residues are incompatible for establishing essential module-module communications since they are originated from different repressors which have evolved under different evolutionary pressures. Allosteric communication appears to play an important role in module compatibility, some previous computational approaches looked at this problem from the perspective of correlated mutations[26–28]. Our work provides evidence that coevolutionary analysis using Direct Coupling Analysis (DCA) is also able to capture, in a global way, this phenomenon. We, however, do not discard the possibility that our analysis also reveals physical residue interactions happening dynamically in conformational ensembles but that have not been observed in known 3D coordinates. To understand the underlying mechanism that governs the activities of hybrid repressors, we used both computational and experimental approaches, in which we first computationally predict residue pairs involved in DBM-LBM interactions based upon coevolutionary traits, which let us to establish a map of evolutionarily dominant module-module interactions among LacI family members; then we compare our established map to the interaction network of hybrid repressors to identify mutations that may restore those lost interactions. Our hypothesis is strongly supported by the fact that we were able to use this approach to rescue four hybrid repressors.

We further investigated why repressor functions of the other four hybrids were not improved with mutations from our compatibility model. Our results suggest that those mutated residues are also involved in intradomain interactions, in which mutating them leads to perturbation of structure and function within the LBM. We introduced a metric for computing a structure-based

score, $SF(\mathbf{S})$, which is a powerful tool for guiding our protein design. Using both the compatibility score $C(\mathbf{S})$ and the structural score $SF(\mathbf{S})$, we were able to enhance our capability to build modular repressors by advancing the understanding on the residue interacting network of LacI family repressors. The combination of these two evolutionary-based metrics allowed us to find the right candidates to improve function and also to discard those that would also be affected within the domain. Our experimental validation shows that our approach is applicable to all 8 hybrids. In the future, we plan to use the combination of $C(\mathbf{S})$ and $SF(\mathbf{S})$ score to design and rescue hybrid repressors with distinct combinations of DBMs and LBMs from the LacI family, continually expanding the toolset and further testing the robustness of this approach. We believe this is an important step towards the rational design of proteins.

Our computational method may open a range of unique possibilities. In addition to rescuing hybrid repressors, it can potentially be used to modulate the behavior of native and hybrid repressors that are functional. There are a number of ways to adjust the dynamic range of inducible gene expressions, such as changing the operator sequence and the strength of the ribosomal binding site and promoter. Programming DBM-LBM interactions provides additional means to further refine induction levels. Furthermore, our modeling strategy should be applicable for studying and engineering repressors in other protein families. We previously developed hybrid repressors from the TetR family and there are similar challenges, including incompatibility of residues for LBM-DBM interactions and infeasibility on modifying residues at the module-module interface as they are also involved in other molecular roles[29]. By applying our approach developed here, we expect to make significant progress in engineering biosensors from a board range of repressor families.

## Methods

**Chemicals and reagents**. All DNA oligonucleotides were synthesized by Eurofins Genomics (Louisville, KY, USA). All enzymes and reagents for cloning were obtained from New England Biolabs (Ipswich, MA, USA). LB medium broth, inducers, and antibiotics were obtained from VWR (Radnor, PA, USA).

**Mutation of hybrid repressors**. A site-directed mutagenesis method is used to mutate hybrid repressor genes[30]. Briefly, a pair of complementary primers were used for PCR to replicate an entire plasmid that contains the hybrid repressor gene. These primers are about 50 base pairs long with target mutation at the center (a list of primers and their sequences are shown in Supplementary Data 1). The hybrid repressor gene-containing plasmid, pTR, was described in our previous publication[9]. PCR products were treated with DpnI to remove template DNA. The resulting PCR product was transformed into *E. coli* XL-1 Blue cells (Agilent Technologies, Inc.; Santa Clara, CA) and selected with LB agar plate that contains 50 ng/mL of kanamycin. Sequences of mutated genes were confirmed by Sanger sequencing with primers listed in Supplementary Data 1.

**Characterization of mutants**. Repressor mutants were characterized by using an in vivo transcriptional assay[9]. The plasmid hosting the repressor (pTR) was transformed into *E. coli* MG1655 with $\Delta lacI$, $\Delta galR$, $\Delta rbsR$, and $\Delta purR$. This strain of cells also contained another plasmid (pREPORT) that has a *GFP* gene; the *GFP* gene was driven by a $P_L$ promoter that contains the desirable operator for interacting with the DBM of the target repressor protein. These cells were grown in LB cultural medium at 37 °C and 200 rpm. At OD600 = 0.3, they were exposed to the effector molecule according to the LBM of the repressor; i.e., repressors containing an LBM originated from LacI, GalR, CelR, ScrR, or RbsR were exposed to IPTG (1 mM), galactose (5 mM), cellobiose (2.5 mM), fructose (5 mM), or ribose (5 mM), respectively. After 3 h, GFP fluorescence of effector-exposed and unexposed cells from each strain was determined by using an ACEA NovoCyte 2030YB flow cytometer (ACEA Biosciences, Inc.). The flow cytometer was set to stop collecting data after acquiring 50,000 events. These data were then gated by forward and side scatter to eliminate multi-cell aggregates, which typically excluded less than 10% of the total events (an example to illustrate our gating strategy is provided in Supplementary Fig. 6), and the geometric means of GFP fluorescence distributions were calculated using the software, NovoExpress 1.4.1 (ACEA Biosciences, Inc.).

This sample size is sufficient for assessing GFP expression as 10,000 events are enough to generate a smooth normal distribution curve for accurately determining the mean value, as shown in Supplementary Fig. 2. The fold induction was reported as the ratio between geometric mean of fluorescence of exposed cells to unexposed cells (Source Data file).

**Coevolution based compatibility score model**. A compatibility score $C(\mathbf{S})$ model inferred from multiple sequence alignments (MSAs) of original repressors from the LacI family was used to predict functionality for hybrid repressors[9]. We've created an MSAs dataset containing 74,287 homologous LacI sequences and it provided 22,090 effective sequences for the model to learn coevolutionary information. The $C(\mathbf{S})$ model was built based on direct coupling analysis (DCA)[22], a method that identifies coevolving residues within protein families. DCA assumes a global statistical model (Boltzmann distribution) for the joint probability distribution of protein family sequences. Therefore, for any given protein sequence vector, $\mathbf{S}$, the probability of occurrence is modeled with pairwise couplings $e_{ij}(A_i, A_j)$ and local biases $h_i(A_i)$.

$$P(\mathbf{S}) = \frac{1}{Z}\exp\left\{\sum_{i<j}e_{ij}\left(A_i, A_j\right) + \sum_i h_i\left(A_i\right)\right\} \qquad (1)$$

Both sets of parameters, $e_{ij}(A_i, A_j)$ and $h_i(A_i)$ are inferred from MSAs of the LacI repressor family with $i$ and $j$ representing positions in the alignment, $A$ is the amino acid identity, and $Z$ is the partition function of $P(\mathbf{S})$. Using the top 1500 coevolving inter-module residue pairs inferred by DCA, the score $C(\mathbf{S})$ uses the coevolutionary coupling strength $e_{ij}(A_i, A_j)$ to estimate the compatibility between DBM and LBM modules for any hybrid repressor with sequence ($\mathbf{S}$) and is defined by the following equation:

$$C(\mathbf{S}) = -\sum_i^{\text{DBM}}\sum_j^{\text{LBM}}\left\{e_{ij}\left(A_i, A_j\right)\right\} \qquad (2)$$

Introducing an amino acid substitution into the hybrid sequence, an updated $C(\mathbf{S})$ score can be calculated to reflect the effect of the mutation on inter-modular compatibility. In this context, a more negative score relative to the score of the original hybrid suggests that this amino acid is more favorable to coevolutionary compatibility between the two domains. Conversely, a more positive score might lead to unfavorable interactions.

**Coevolution based structural fitness model**. In order to avoid mutations that will affect the folding or stability of the hybrid protein, an additional scoring model is proposed as $C(\mathbf{S})$ focuses on the effect on inter-module interaction and does not incorporate intra-module information. To achieve this, coevolutionary coupling strength parameters $e_{ij}(A_i, A_j)$ for residues that are in physical contact (<10 Å) in the experimental structure of the native domains (PDB: 1lbg) were used to account for the mutational effect on the structural stability or function. When a mutation occurs on a residue in contact with others and disrupted the coevolutionary constraints with those residues, a less negative score relative to the original score would be expected. In contrast, a more negative score is suggestive of a favorable effect. The following equation quantifies the evolutionary interactions happening at each module:

$$SF(\mathbf{S}) = -\sum_{i,j}^{\text{Contacts}}\left\{e_{ij}\left(A_i, A_j\right)\right\} \qquad (3)$$

It is important to note that the DBM and the LBM have very limited physical interactions and a relevant part of the communication appears to be allosteric. Therefore, the $SF(\mathbf{S})$ score only affects the independent folding units of the DBM and LBM.

**Reporting summary**. Further information on research design is available in the Nature Research Reporting Summary linked to this article.

## Data availability
All data for the reproduction of the computational result have been deposited in a GitHub repository: https://github.com/morcoslab/coevolution-compatibility, including LacI homologous MSAs collected from Uniprot database[31], and physical contact residues extracted from a crystal structure of LacI from PDB database under accession code 1LBG[32]. Source data are provided with this paper. All hybrid repressor characterization data generated in this study are provided in the Source Data file. Source data are provided with this paper.

## Code availability
The collection and alignment of LacI homologous sequences was conducted utilizing HMMER 3.2.1[33] on sequences from the Uniprot database[31]. All codes that perform the DCA method and $C(\mathbf{S})$/$SF(\mathbf{S})$ score calculations for this article are written in MATLAB R2019b. The raw codes, LacI family MSAs dataset for the coevolutionary model, aligned sequence files for all 8 original repressors as well as the parameter files used for this study can be accessed at GitHub: https://github.com/morcoslab/coevolution-compatibility and on Zenodo (https://doi.org/10.5281/zenodo.5262799)[34].

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

## Acknowledgements

This work was supported by funding from the National Institutes of Health/National Institute of General Medical Sciences grants 1R15GM135813-01 (C.T.Y.C.), R35GM133631 (X.J. and F.M.), and the National Science Foundation grant MCB-1943442 (F.M.).

## Author contributions

X.J., R.P.D., C.T.Y.C. and F.M. designed the study, analyzed data, and wrote the paper. X.J. and F.M. performed the computational modeling studies. R.P.D. and C.T.Y.C. performed biological experiments.

## Competing interests

The authors declare no competing interests.
