## [Peer Review File · Nature Communications]

Reviewers' Comments:

Reviewer #1:

Remarks to the Author:

The authors describe a rational protein design strategy for engineering hybrid repressors consisting of various ligand-binding and DNA-binding modules (LBMs and DBMs) within the LacI family. Fusing LBMs to DBMs from different repressors introduces novel pathways between specific ligands and regulation of specific promoters that are useful for genetic circuit design. Hybrid repressors often function poorly, due to the loss of critical LBM-DBM interaction. The authors use a computational model to identify co-evolved residues important in LBM/DBM interaction combined with structural considerations enable the authors to make distinct point mutations that significantly improve the functionality of hybrid repressors. Hybrid repressors are useful for genetic circuit design, sensor screening, or bioproduction. However, high-throughput library screening is resource-intensive. This approach could significantly improve the success rate of hybrid transcriptional regulators, even outside the LacI family.

Comments:

1. As a whole, this is a well-organized study with clear motivation, approach, and experimentation.
2. The authors should discuss in more detail why only 8 hybrids were selected from the previous study- how were hybrids with the "majority of essential LBM-DBM interactions" intact identified?
3. The C(S) and SF(S) metrics appear to be predictive but a more detailed assessment would be useful. Plots relating fold change to both C(S) or SF(S) would reveal how predictive these metrics are. Ideally, these could include data from this study as well as the previous study (ref 9), which covered successful hybrids that required no additional mutation.
4. The authors should discuss whether or not this approach could be applied to the previously successful hybrids (ref 9). Did they already happen to have the maximum possible C(S) and SF(S) scores?
5. It would be beneficial to include a generic LacI structure, preferably with DNA and dimeric partners included, with important residues labelled such as the top 5 single point mutation positions so that the reader can see where these mutations physically are rather than just sequence position.
6. In general, more detailed discussion on the structure and mechanism of LacI-type regulators would be appropriate, particularly related to the unsuccessfully-recovered hybrids. LacI is well-known to form dimers and tetramers, but this property was not discussed in the manuscript.
7. In figure 1, the authors should use the same colorbar and C(S) bounds across all panels so the favorability of single/double/triple mutations are comparable.
8. In SF1, the authors should use the same colorbar and C(S) bounds across all panels so the magnitude of $\Delta C(S)$ is comparable across all hybrids, rather than relative comparisons within each panel.
9. In SF1, the authors should include LacI-RbsR and stack identical LBMs vertically so that trends in favorable mutation position are visible (xxxx-RbsR in panels a/c, xxxx-GalR in b/d, etc.)
10. In figure 1, the authors should use consistent position bounds for all axes in all panels
11. In figure 1 and SF1, it would be beneficial to highlight the best mutants (small boxes or circles) for easier identification.
12. In figure 1, consider using the same axes for panels c and d, such that in c all points lie on a flat plane. It is a little bit confusing to see C(S) as both an axis and colorbar in panel c while it is just a colorbar in b and d.
13. I like the presentation of data in Figure 2 showing C(S), original, and mutated performance. Consider plotting individual cell fluorescence distributions +/- ligand. It would be interesting to see if all populations are normally distributed.
14. It is mentioned that the favorable mutations for xxxx-RbsR hybrids are "in the binding groove with DNA". Please specify if this position actually contacts DNA or is just DBM-adjacent.
15. The section title "Folding and structural constraints on high compatible mutants" isn't clear. Do the authors mean to say "highly compatible mutants"?
16. Please discuss if it is possible to modify the SF(S) score to introduce a penalty for residue proximity to ligand binding pocket.
17. In figure 3, the authors should adjust C(S) and SF(S) axes bounds to be identical across all panels so it is easier to compare the magnitude of score change across different hybrids and their variants.

18. It would be useful to plot Δ fold change as a function of both $\Delta C(S)$ and $\Delta SF(S)$ across all variants, to better assess how well they predict advantageous mutations. It may be necessary to plot as normalized percentages relative to original hybrid, given the varying scales of scores and fold changes. For example, LacI-RbsR(K57A) exhibits a $\sim 4400\%$ increase in fold change across a $\sim 6.7\%$ change in $C(S)$ and $\sim 0.1\%$ change in $SF(S)$

Reviewer #2:

Remarks to the Author:

In the manuscript "Coevolutionary methods enable robust design of modular repressors by reestablishing intra-protein interactions", the authors use a DCA-based computational model, which was previously developed to identify critical residues involved in interactions between DNA-binding modules and ligand-binding modules, to determine mutations that can restore the module-module interactions and rescue the hybrid repressors. They also study the phenotypic effect of the proposed critical mutations and complemented the model further by incorporating structural information.

Overall, this work is quite interesting as it shows the potential use of computational models for making meaningful predictions that are backed up by experiments. However, there are some parts of the paper that need further elaboration (see below). With my expertise in the computational and statistical side, my comments are mostly about the methodological details. I would leave the review of experimental part to the experts in that area.

1. How is this work different from inferring interaction partners work of Bitbol et al (10.1073/pnas.1606762113)? Similar approach, i.e., based on DCA formulation. But I understand that the objective is different. In this work, Bitbol et al. try to identify partners while the current work tries to identify mutations possibly responsible for the interaction between partners.

Can a similar approach be used to identify which LBM and DBM may interact with each other to form functional hybrids? and in the process identify new hybrid repressors? A discussion related to this paper and other related papers would be enlightening for the readers.

2. A heatmap is shown in Fig. 1B to illustrate the effect on compatibility score of all possible single mutations for this hybrid repressor. Among the mutations that improve the compatibility score, the top 5 favorable mutations are ...

How much are couplings playing a role? Can these mutations be predicted using conservation/single site mutations?

How many sequences were there? How rich was the data? Regularization?

3. Discussion, paragraph 2 and related results:

While interesting, this approach may be just applicable to the current specific mutations that the authors investigated. To test if the approach is more general, authors could have used the combination of $C(S)$ and $SF(S)$ to make similar further predictions.

4. It is important to note that the DBM and the LBM have very limited physical interactions and most of the communication appears to be allosteric. Therefore, the $SF(S)$ score only affects the independent folding units of the DBM and LBM

To the reviewer's understanding, DCA is good in inferring protein physical contacts. However, is there any study that indicates DCA predicts allosteric interactions as well? On the other hand, methods based on correlated mutations (such as 10.1038/ncomms4287; 10.1038/msb.2010.65; and 10.1371/journal.pcbi.1006409) have been shown to predict allosteric interactions in proteins. A discussion about this related research would be helpful.

5. In this context, a more negative new score suggests that this amino acid is more favorable to

coevolutionary compatibility between the two domains...

Shouldn't this score be computed with respect to some reference? A "more negative score" than what? This looks confusing.

Minor comments:

6. What is S in C(S) on page 5? This is not explained until the Methods section.

7. Using this computational model, we systematically computed C(S) scores for mutations at LBMs.

8. Shouldn't this C be a function of j then? Confusing statement.

9. We did not consider mutations at DBMs because these modules are small (approximately 47 amino acid residues) and many residues are directly involved in DNA binding and recognition; mutating a DBM is likely to affect DNA binding properties of the protein

Reference for this?

10. We then selected mutants with the best C(S) scores, ...

Mutants of LBMs?

Dear reviewers,

Thank you for your constructive comments and suggestions. We have addressed all your points and provided a description of our responses below. We made comprehensive changes in the presentation and content of our manuscript, including revised Figs. 1, 2, and 3, as well as several new Supplementary figures and accompanying changes to the manuscript. In addition to the changes and edits made in response to your inquiries, we also made modifications that satisfy the requirements from *Nature* journals to the presentation and reporting of flow cytometry data. This is reflected in new Supplementary Figures 2 and 6. We are confident that this new version of our manuscript is improved in clarity and content, and is a better description of the results presented in the original manuscript. The inquiries and responses are shown in emboldened and green font, respectively. We have also included a new version of the manuscript with the changes marked in red.

RESPONSE TO THE REVIEWERS' COMMENTS

Reviewer #1 (Remarks to the Author):

The authors describe a rational protein design strategy for engineering hybrid repressors consisting of various ligand-binding and DNA-binding modules (LBMs and DBMs) within the LacI family. Fusing LBMs to DBMs from different repressors introduces novel pathways between specific ligands and regulation of specific promoters that are useful for genetic circuit design. Hybrid repressors often function poorly, due to the loss of critical LBM-DBM interaction. The authors use a computational model to identify co-evolved residues important in LBM/DBM interaction combined with structural considerations enable the authors to make distinct point mutations that significantly improve the functionality of hybrid repressors. Hybrid repressors are useful for genetic circuit design, sensor screening, or bioproduction. However, high-throughput library screening is resource-intensive. This approach could significantly improve the success rate of hybrid transcriptional regulators, even outside the LacI family.

Comments:

1. As a whole, this is a well-organized study with clear motivation, approach, and experimentation.

We would like to thank this reviewer for this encouraging comment and for spending time reviewing our manuscript.

2. The authors should discuss in more detail why only 8 hybrids were selected from the previous study- how were hybrids with the “majority of essential LBM-DBM interactions” intact identified?

We would like to thank this reviewer for raising this question. In our previous study (PMID: 31162606), we used 5 LBMs and 8 DBMs to construct and characterize a total of 35 hybrid repressors and 5 native repressors. Among the 35 hybrids, 18 of them are highly efficient (fold-change of GFP expression >10 upon induction), 8 hybrids have reduced activities (fold-change is between 3 and 10) and 7 hybrids have no observable activities (fold-change less than 3).

Our results strongly support that individual LBMs and DBMs retain the original ligand-sensing and DNA recognition properties, respectively, because all of them can be used to create efficient hybrids, which led us to hypothesize that reduced activities in the 8 hybrids (and also no activities in the other 7 hybrids) are due to interrupted interactions between the LBM and DBM. For the 8 cases with reduced activities, the fact that these hybrids still have some biological activities suggest that the LBM and the DBM can still interact to some extent; however, for those other 7 hybrids with no activities, all module-module interactions might be

completely lost. Rescuing the 7 no-activity hybrids may require more mutations to rebuild all the module-module interactions, compared to rescuing the 8 reduced activity hybrids, and with more mutations, there is a higher chance to ruin other intramodular interactions, rendering more challenging to get an efficient mutated hybrid. Therefore, as a first step to assess whether we can utilize coevolutionary traits to reestablish module-module interactions, we chose to apply our strategy to those 8 hybrids with reduced activities, which we believe are more promising to rescue them with our approach.

We have elaborated the paragraph started on **page 6** to describe our rationale on selecting hybrid repressors for this study.

3. The C(S) and SF(S) metrics appear to be predictive but a more detailed assessment would be useful. Plots relating fold change to both C(S) or SF(S) would reveal how predictive these metrics are. Ideally, these could include data from this study as well as the previous study (ref 9), which covered successful hybrids that required no additional mutation.

We thank the reviewer for suggesting a clearer way to present our plots. We agree that plots that contain fold change information along with C(S) and SF(S) would be more informative. Therefore, we have made changes in **Figure 3** by integrating a new colorbar that indicates the fold change values for each mutant relative to its original hybrid. The new plots are easier to read and illustrate in a better way how a combination of both C(S) and SF(S) is a better predictor of functional restoration. Particularly, the SF(S) serves as a constraint that needs to be achieved in order to improve the fold change response. We decided not to include data from our previous study as ref. 9 already demonstrates a strong predictive value (AUC value of 0.88) for original hybrids; since our current study deals with relative changes in the score among mutants from the same original hybrid, we conclude that including previous results from other hybrid repressors would be confusing to the reader of this present study. The new **Figure 3** condenses in a better way the relationship between the C(S), SF(S), and the experimentally observed fold changes.

4. The authors should discuss whether or not this approach could be applied to the previously successful hybrids (ref 9). Did they already happen to have the maximum possible C(S) and SF(S) scores?

Thanks for bringing up this interesting idea. When we first decided to work on this project, we thought that focusing on rescuing repressors with poor activities was a more challenging task in comparison to optimizing repressors with high induction fold-change. We reasoned that making previously incompatible mutants into compatible ones would lead to stronger results versus focusing on functional ones. Having said that, we believe there is still room for improvement in successful hybrids. From our experience, repressors that can generate >20 fold of induction are promising to serve as parts for building genetic circuits. Conventionally, a number of methods are used to adjust the dynamic range of inducible expression systems in synthetic biology, including changing ribosomal binding site, plasmid copy number, expression levels of the repressor, the number of operators for the repressor to bind, and promoter strength. Our computational design to modulate DBM-LBM interactions can be a new way to program induction response. We have highlighted this potential application in the last paragraph of the Discussion section (**pages 14 and 15**).

5. It would be beneficial to include a generic LacI structure, preferably with DNA and dimeric partners included, with important residues labelled such as the top 5 single point mutation positions so that the reader can see where these mutations physically are rather than just sequence position.

We also think that this is a good suggestion as a crystal structure can help readers to visualize the position of these mutations. We have included a new **Supplementary Fig. 3** as suggested by this reviewer, which includes a homodimer of native LacI. On the LacI structure, we also indicate the homologous positions in the rescued hybrids, which are mutated to reach the best performance. This Supplementary Figure is discussed in the main text on **pages 8 and 9**.

6. In general, more detailed discussion on the structure and mechanism of LacI-type regulators would be appropriate, particularly related to the unsuccessfully-recovered hybrids. LacI is well-known to form dimers and tetramers, but this property was not discussed in the manuscript.

In the revised manuscript, we have used RbsR-LacI as an example to discuss how the mutations may affect protein structure and function, which lead to unsuccessfully-recovered hybrids (**page 12, last paragraph in the Results section**). We use our SF(S) model to identify potential disrupted interactions between the mutation and other residues, which helps us to rationalize why these mutants are unsuccessful. These analyses show a potential way of using the SF(S) model to understand the molecular mechanism of these proteins. A new **Supplementary Fig. 5** is added to illustrate our ideas.

7. In figure 1, the authors should use the same colorbar and C(S) bounds across all panels so the favorability of single/double/triple mutations are comparable.

We agree that a better scheme to compare single/double/triple mutants is needed for figure 1. Instead of using a uniform colorbar for all three panels, we used $\Delta C(S)$ relative to the score for original repressor (**Figure 1b**), the best single mutation score (**Figure 1c**) and the best double mutation score (**Figure 1d**). We think this better reflects the actual design and experimental process where we first identify the best candidate, test it experimentally and then we proceed with the best relative score for the double mutant. This new scheme is easier to read as each part can be read independently and just relative to the previous score which allows higher resolution when identifying and displaying the best candidates to test.

8. In SF1, the authors should use the same colorbar and C(S) bounds across all panels so the magnitude of $\Delta C(S)$ is comparable across all hybrids, rather than relative comparisons within each panel.

We've made corresponding changes in **Supplementary Figure 1**. To allow the use of the same colorbar for all 8 hybrid repressors and to make it consistent with the new **Figure 1**, we plotted $\Delta C(S)$ scores in the heatmap and labeled the original score in the title of each panel. We believe this improves readability and thank the reviewer for this suggestion.

9. In SF1, the authors should include LacI-RbsR and stack identical LBMs vertically so that trends in favorable mutation position are visible (xxxx-RbsR in panels a/c, xxxx-GalR in b/d, etc.)

We've made the corresponding changes suggested by the reviewer in **Supplementary Figure 1**. Additionally, all x-axis bounds are changed to be the same. We also re-arranged the order of hybrid repressors in **Figure 3** to keep consistent with Supplementary Fig. 1.

10. In figure 1, the authors should use consistent position bounds for all axes in all panels

Thanks for the suggestion, we've corrected the position bounds in the new figure.

11. In figure 1 and SF1, it would be beneficial to highlight the best mutants (small boxes or circles) for easier identification.

The best 5 single mutants in **Figure 1** and **Supplementary Figure 1** were highlighted with arrows; For the best ones in Figure 1b, c, we highlighted the best 3 with a large symbol.

12. In figure 1, consider using the same axes for panels c and d, such that in c all points lie on a flat plane. It is a little bit confusing to see C(S) as both an axis and colorbar in panel c while it is just a colorbar in b and d.

Thanks for pointing out this confusing part. We've used a 2D plot to represent the data in **Figure 1c**, so that x and y axes indicate the residue positions, and the C(S) score is represented with a colormap to keep it consistent with panels **b** and **d**.

13. I like the presentation of data in Figure 2 showing C(S), original, and mutated performance. Consider plotting individual cell fluorescence distributions +/- ligand. It would be interesting to see if all populations are normally distributed.

Thank you for reminding us to illustrate the raw data acquired with the flow cytometer. We have now shown those data in **Supplementary Fig. 2**, which includes all three replicates for each experiment. These illustrations show that GFP fluorescence of all experiments is normally distributed and is highly reproducible among the three biological replicates.

14. It is mentioned that the favorable mutations for xxxx-RbsR hybrids are “in the binding groove with DNA”. Please specify if this position actually contacts DNA or is just DBM-adjacent.

This position (K57 in LacI-RbsR and K60 in CelR-RbsR) is at the hinge helix motif, which interacts with the backbone of DNA but not the nucleobases. Thus, in our previous study (PMID: 26641934), we proposed that it is involved in facilitating the protein-DNA binding but not the recognition of DNA sequence. Based on crystal structures, this position is in physical contact with DNA. Intriguingly, our model shows that this position has strong coevolution activities with the DBM. We think that with different amino acid residues, this position interacts with specific parts of the DNA backbone (such as the ribose and the phosphate group), such that the DNA and DBM are at a desirable orientation to form a complex. We have further the discussion on this point in the subsection “Designing hybrid repressors to improve allosteric regulation activities” on **page 9** in the new version of the manuscript. We thank the reviewer for bringing this up.

15. The section title “Folding and structural constraints on high compatible mutants” isn't clear. Do the authors mean to say “highly compatible mutants”?

Thank you for catching this typo. We have corrected that on **page 10**.

16. Please discuss if it is possible to modify the SF(S) score to introduce a penalty for residue proximity to ligand binding pocket.

As shown in **Figure 3**, there are three SF(S) score outliers that are potentially due to interactions between two mutated residues and the ligand. We may solve this problem by prohibiting mutations at the ligand-binding pocket. In our current model, we do not consider any mutations in the DNA-binding module (residues 1 to 47 of LacI) because many residues there are highly involved in interacting with the DNA operator. Similarly, we could penalize mutants that involve changes in the ligand-binding pocket. We have discussed that on the subsection “Folding and structural constraints on highly compatible mutants” on **page 12**, in the revision of this manuscript. We plan to incorporate this feature in future iterations of the model. We thank the reviewer for this suggestion.

17. In figure 3, the authors should adjust C(S) and SF(S) axes bounds to be identical

across all panels so it is easier to compare the magnitude of score change across different hybrids and their variants.

As shown in the new **Figure 3**, all the C(S) and SF(S) axes bounds were changed to be the same. We agree this makes the data presentation clearer and more accessible to the reader.

18. It would be useful to plot Δ fold change as a function of both Δ C(S) and Δ SF(S) across all variants, to better assess how well they predict advantageous mutations. It may be necessary to plot as normalized percentages relative to original hybrid, given the varying scales of scores and fold changes. For example, LacI-RbsR(K57A) exhibits a ~4400% increase in fold change across a ~6.7% change in C(S) and ~.1% change in SF(S)

In the new **Figure 3**, we have plotted the fold change relative to the original hybrid (ratio of mutant score to original score) along with both Δ C(S) and Δ SF(S) scores to facilitate the analysis of the data. We decided to present the fold change in a log scale instead of normalized percentages. It is important to mention, that the change in C(S) and SF(S) scores represent changes of parameters in an exponential probability distribution, therefore small changes in the scores could have a non-linear relationship with the fold changes.

Reviewer #2 (Remarks to the Author):

In the manuscript “Coevolutionary methods enable robust design of modular repressors by reestablishing intra-protein interactions”, the authors use a DCA-based computational model, which was previously developed to identify critical residues involved in interactions between DNA-binding modules and ligand-binding modules, to determine mutations that can restore the module-module interactions and rescue the hybrid repressors. They also study the phenotypic effect of the proposed critical mutations and complemented the model further by incorporating structural information.

Overall, this work is quite interesting as it shows the potential use of computational models for making meaningful predictions that are backed up by experiments. However, there are some parts of the paper that need further elaboration (see below). With my expertise in the computational and statistical side, my comments are mostly about the methodological details. I would leave the review of experimental part to the experts in that area.

We would like to thank the reviewer for the encouraging assessment of our work and its results and for providing feedback to make the manuscript clearer and more detailed. Please find below the answers to all of the reviewer’s comments.

1. How is this work different from inferring interaction partners work of Bitbol et al (10.1073/pnas.1606762113)? Similar approach, i.e., based on DCA formulation. But I understand that the objective is different. In this work, Bitbol et al. try to identify partners while the current work tries to identify mutations possibly responsible for the interaction between partners.

Can a similar approach be used to identify which LBM and DBM may interact with each other to form functional hybrids? and in the process identify new hybrid repressors? A discussion related to this paper and other related papers would be enlightening for the readers.

We thank the reviewer for bringing up this relevant work. In fact, the conceptual basis between identifying interacting specificity among independently folding units using a metric of

coevolution is similar. In Bitbol et al. an algorithmic approach to iteratively optimize coevolutionary couplings is able to identify specific interacting partners, e.g. in Histidine Kinase – Response regulator two-component systems. The current work uses the same principle to quantify evolutionary compatibility between units. In this case, the units are covalently bonded therefore a methodology to identify native compatibility is not crucial as the domains are part of the same repressor protein. In addition, to this, the current work not only identifies potential interactors but introduces a design process to systematically identify such interactions that might restore compatibility by taking into account both interdomain interactions as well as changes that might disrupt the single domain stability. Another major difference is that in our work we not only describe interactions between native sequences but proposes changes that introduce non-native sequences that improve repressor activity guided by the coevolutionary considerations. We also confirm experimentally such designed sequences and provide a strong proof of concept that this methodology can be used to create libraries of repressors with multiple applications in synthetic biology.

We think that the use of IPA could be interesting in future work where we would like to design interacting networks domains that are not necessarily bonded. We believe an approach like that would be interesting and relevant for future studies. In the new text we have made a reference to the work of Bitbol et al. and other related works (**page 5**) and highlighted how it serves as conceptual support to the motivation for our work.

2. A heatmap is shown in Fig. 1B to illustrate the effect on compatibility score of all possible single mutations for this hybrid repressor. Among the mutations that improve the compatibility score, the top 5 favorable mutations are ...

How much are couplings playing a role? Can these mutations be predicted using conservation/single site mutations?

How many sequences were there? How rich was the data? Regularization?

Only couplings are used in the calculation of the C(S) score. Conservation could provide limited predictions here. Residue 57, 75, and 163 and 293 in this case were not conserved sites. If we consider residue 57 as an example, the most conserved amino acids were 'R', 'A', 'K', 'V' in descending order. If we rely only on conservation, we would predict K57R be the best mutant when in fact it is K57A or K57V (**Figure 1b**). Another example would be A87C in RbsR-GalR and homologous A85C in PurR-GalR, where A is the most conserved amino acid at 87, however A87C (A85C), improves protein function. This provides further support of the importance of epistatic interactions in designing novel sequences with desired properties.

Our MSAs dataset is abundant and contains 74,287 homologous sequences, with 22,090 effective sequences after reweighting at 0.8 identity. We've also included this information in the Methods section in the revised version of the manuscript (**page 19**).

3. Discussion, paragraph 2 and related results:

While interesting, this approach may be just applicable to the current specific mutations that the authors investigated. To test if the approach is more general, authors could have used the combination of C(S) and SF(S) to make similar further predictions.

We strongly agree that experimentally testing our approach with additional hybrids can provide stronger support on the robustness and generality of our approach for rescuing hybrid repressors. As a first step, we have tested the approach on 8 hybrids (11 mutants for each hybrid), involving a total of 88 mutants and the 8 original hybrid repressors. With this reviewer's suggestion, we have highlighted in the Discussion section the importance of further testing our approach (**page 14**). Hopefully, publishing this article will help us to gather more resources to continue our experimental work for this project.

4. It is important to note that the DBM and the LBM have very limited physical interactions and most of the communication appears to be allosteric. Therefore, the SF(S) score only affects the independent folding units of the DBM and LBM

To the reviewer's understanding, DCA is good in inferring protein physical contacts. However, is there any study that indicates DCA predicts allosteric interactions as well? On the other hand, methods based on correlated mutations (such as 10.1038/ncomms4287; 10.1038/msb.2010.65; and 10.1371/journal.pcbi.1006409) have been shown to predict allosteric interactions in proteins. A discussion about this related research would be helpful.

DCA has been successful inferring residue-residue contacts, however, it has also been used to predict other global metrics of protein folding and function like specificity (Cheng et al. PNAS, 2014. doi.org/10.1073/pnas.1323734111), Phi values (Cheng et al. Protein Science, 2015. <https://doi.org/10.1002/pro.2758>), mutational effects (Figliuzzi et al. MBE, 2016. 10.1093/molbev/msv211), selection temperatures (Morcos et al. PNAS, 2014 doi.org/10.1073/pnas.1413575111), folding rates (Malik et al. PEBS letters, 2015. doi.org/10.1016/j.febslet.2015.06.032), among several other biologically relevant order parameters and not only physical contacts. The case of allostery has also been investigated via DCA (Baldessari et al. CSBJ, 2020. doi.org/10.1016/j.csbj.2020.05.003), including our previous paper (Dimas et al. NAR, 2018. doi.org/10.1093/nar/gkz280) that describes compatibility between repressor domains. Other metrics of correlated mutations, like Mutual Information might also be informative on allosteric mechanisms. In this case, the global nature of the probabilistic model captured in the DCA Hamiltonian seems to capture relevant interactions even if they are not necessarily in physical contact. Having said that, the reviewer brings an important point, that merits clarification in our manuscript. Although, the contacts are limited between the two repressor domains, it does not necessarily mean that those contacts are irrelevant. In addition to that, since not all the conformational dynamics of these repressors have been unraveled, there is still a possibility that some of the interactions that we find important, although distant in the known crystals might be, in fact, physically interacting in alternative conformations of the proteins. Therefore, we modified our manuscript to provide more mechanistic details and supplementary figures (Supplementary Figures 3 and 5) on the allosteric nature of the domain-domain communication (see section "Designing hybrid repressors to improve allosteric regulation activities"). Although this phenomenon is not completely understood, our work provides evidence that some of its features can be captured by the evolutionarily conserved set of couplings that constitute function in the repressors. We have included a discussion of this in the Discussion section of the manuscript, including some of the references mentioned by the reviewer (**page 13**).

5. In this context, a more negative new score suggests that this amino acid is more favorable to coevolutionary compatibility between the two domains...

Shouldn't this score be computed with respect to some reference? A "more negative score" than what? This looks confusing.

We would like to thank the reviewer for identifying this confusing statement. The reviewer is correct that our scores should be compared against a reference score. In our study, mutant scores are compared against the original hybrid score which we are trying to rescue. Therefore, 'a more negative score' means 'more negative score relative to the score of original hybrids. We've clarified this in the new version of the manuscript (see **page 20**).

Minor comments:

6. What is S in C(S) on page 5? This is not explained until the Methods section.

The S in the C(S) score is a particular sequence for a hybrid. The score is a single number per sequence and changes as mutations are introduced. We agree that an introduction in the methods section is cumbersome and unclear, therefore we have added an explanation of S where it appears first time in the manuscript (**page 5**).

7. Using this computational model, we systematically computed C(S) scores for mutations at LBMs. Shouldn't this C be a function of j then? Confusing statement.

The confusion here will be clarified as now we have introduced the definition of S early in the manuscript. The C(S) score depends on a collection of couplings, it is analogous as the probability P(S) in the entropy maximization procedure in the formulation of DCA, where the probability depends, in a joint way, on the whole sequence. There is only a single value of C(S) for a particular sequence, and it does not depend on *j* or *i* but on the sum of all the couplings in the top ranked pairings as defined in Dimas et al. NAR, 2019 (PMID: 31162606).

9. We did not consider mutations at DBMs because these modules are small (approximately 47 amino acid residues) and many residues are directly involved in DNA binding and recognition; mutating a DBM is likely to affect DNA binding properties of the protein

Reference for this?

In a previous study (P. Markiewicz et al. J. Mol Bio. 1994, PMID: 8046748), 4000 single mutants of LacI were characterized, with each position mutated to 12-13 different amino acids. These mutants show that 28 of the residues in positions 2 to 47 are essential, in which mutating them led to complete loss of DNA binding and loss of function. Since DNA recognition is not modeled in our methodology, we decided to avoid mutating residues in this region. This study is cited in the revised manuscript (**page 6**).

10. We then selected mutants with the best C(S) scores, ...

Mutants of LBMs?

The reviewer is correct, the changes occur at the LBM. We have made it explicit in the new version of the manuscript (**page 6**).

Reviewers' Comments:

Reviewer #1:

Remarks to the Author:

The authors have addressed all of my concerns.

Reviewer #2:

Remarks to the Author:

I would like to thank the authors for addressing all my concerns in such a detailed manner. This is a very nice and interesting work, for which I would like to congratulate the authors. I have no further comments.

Dear reviewers,

Thank you again for your inputs to our manuscript. We truly believe that these inputs are very important; they have led to significant improvement of our work.

RESPONSE TO THE REVIEWERS' COMMENTS

Reviewer #1 (Remarks to the Author):

The authors have addressed all of my concerns.

We are glad to receive your questions and suggestions.

Reviewer #2 (Remarks to the Author):

I would like to thank the authors for addressing all my concerns in such a detailed manner. This is a very nice and interesting work, for which I would like to congratulate the authors. I have no further comments.

Thank you for your encouraging note.